# Burden of tobacco-related cancers in urban, semi-urban and rural setting of Nepal: Findings from population-based cancer registries 2019

Uma Kafle Dahal[1]*, Meghnath Dhimal[1], Atul Budukh[2], Kopila Khadka[1], Sudha Poudel[1], Gehanath Baral[1], Pradip Gyanwali[1], Anjani Kumar Jha[3], Sandhya Chapagain[4]

1 Nepal Health Research Council, Ramshah Path, Kathmandu, Nepal, 2 Centre for Cancer Epidemiology (ACTREC), Tata Memorial Centre, Mumbai, Homi Bhabha National Institute, Mumbai, India, 3 Department of Radiation Oncology, Kathmandu Cancer Center, Tathali, Bhaktapur, Nepal, 4 Department of Clinical Oncology, National Academy of Medical Sciences (NAMS), Bir Hospital, Kathmandu, Nepal

* dahaluma1@gmail.com

**Data Availability Statement:** All relevant data are within the paper.

## Abstract

### Background

Nepal is one of the high prevalent countries for tobacco use in Southeast Asia regions. Tobacco related cancer share the major burden since a decade, however, population-based estimates is still lacking. This study provides results from population-based cancer registries on tobacco-related cancer (TRCs) burden in Nepal.

### Methods

The data were collected by population-based cancer registry conducted in nine districts by Nepal Health Research Council. The districts were categorized in urban, semi-urban and rural regions on the basis of geographical locations and facilities available in the regions. Analysis was done to identify tobacco-associated cancer incidence, mortality and patterns along with cumulative risk of having cancer before the age of 75 years.

### Results

Tobacco-related cancer was 35.3% in men and 17.3% in women. We found that every one in 36 men and one in 65 women developed tobacco-related cancer before age 75 in Nepal. Cancer of lung, mouth, esophagus and larynx were among the five most common tobacco-related cancers in both men and women. The incidence of tobacco-associated cancers was higher in urban region with age adjusted rate 33.6 and 17.0 per 100,000 population for men and women respectively compared to semi-urban and rural regions. Tobacco-associated cancer mortality was significantly higher compared to incidence.

**Funding:** The author(s) received no specific funding for this work.

**Competing interests:** The authors have declared that no competing interests exit.

## Conclusion

The prevalence of tobacco-related cancer found high in Nepal despite of enforcement of tobacco control policy and strategies including WHO framework convention on tobacco control. Concerned authorities should focus towards monitoring of implemented tobacco control policy and strategies.

## Introduction

Globally, the burden of cancer has increased annually. According to GLOBOCAN, 19.3 million new cases and 10 million death of cancer were estimated in the year 2020 which is expected to raised by 47% more in 2040 [1]. World Health Organization (WHO) claims that 30–50% of total global cancer can be prevented through application of existing evidence-based preventive strategies [2]. Carcinogens found in tobacco (smoked and smokeless tobacco) are responsible for causing numerous cancers associated with tobacco use, and tobacco control and lifestyle modification are the scientifically established preventable measures for cancer and also promotes cancer survival without complications [3]. Based on the Global Cancer Observatory (GLOBOCAN) estimates for Nepal, more than twenty thousand (20805) new cancer cases and more than thirteen thousand (13,629) deaths were reported in 2020 [4]. As reported by latest results of Nepal burden of disease survey, malignant neoplasm shared 11% of total deaths among NCDs [5].

Tobacco use including smoking is one of major single preventable cause of death in the world [3, 6]. WHO Framework Convention on Tobacco Control (WHO FCTC) is the first treaty negotiated in response to the globalization of tobacco epidemics. To reduce demand for tobacco use, WHO FCTC endorsed some measures regarding protection from exposure to tobacco smoke, tobacco contents regulation and its disclosure, packaging and labeling, education, communication, training and awareness, and tobacco advertising and sponsorship etc. Nepal had initiated tobacco control and prevention through endorsement of policies since 1990s, and the major intervention to control tobacco were taxation on international brands for sales and distribution of tobacco products and banning of tobacco advertisement on audio-visual media [7]. Nepal signed the WHO FCTC on 3rd December 2003 which was ratified from the house of representative on 7th November 2006. Nepal has enacted the Tobacco Product Control Act 2011 which has incorporated key provision of WHO FCTC and MPOWERs measures [8]. MPOWER is an aggregated term for controlling tobacco product and it's consumption where each words stands for Monitor tobacco use and Prevention policy, Protect people from tobacco smoke, Offer help to quit tobacco smoking, Warn about the dangers of tobacco, Enforce ban on tobacco advertising, promotion and sponsorship and Raise taxes on tobacco, respectively. Moreover, tobacco demand reduction strategies were implemented with high priority through Nepal Health Sector Program (2010–2015) and the SDGs (2016–2030) has addressed to effectively implement the WHO FCTC to reduce prevalence of tobacco user in Nepal as well [8].

According to the results of global smoking epidemics, 32.6% and 6.5% adult men and women respectively smoke globally; the prevalence of smoking has dropped considerably in countries with higher socio-demographic countries. Some countries such as Nepal, the Netherlands, and Denmark showed substantially declined in smoking prevalence in women since 1990, however, overall cigarettes consumption is still growing globally [9]. According to STEP wise approach to NCD risk factors surveillance in Nepal (2019), the prevalence of current

tobacco user was 28.9% (48.3% men and 11.6% women) among them 17.1% (28.0% men and 7.5% women) used tobacco smoking and 18.3% (33.3% men and 4.9% women) used smokeless tobacco product [10].

Tobacco contains more than 4000 chemicals including more than 60 known carcinogen in cigarette smoke and at least 16 in smokeless tobacco [6]. Tobacco use has been found significantly associated with the development of cancer in most part of the human body however, oral cavity, pharynx, larynx, lung, esophagus, pancreas, urinary bladder, and kidney cancer have been recognized as major tobacco-related cancers (TRCs) [6, 11]. Cigarette smoking is responsible for 90% of lung cancer which is the major cause of cancer related death globally. Similarly, smokeless tobacco use has been identified as anticipated risk factors for several cancers including oral cavity pharynx, larynx, esophagus, lung, pancreas and many more. Tobacco control (tobacco smoking as well as smokeless tobacco) is the scientifically proven single most preventable cause of death [6, 30].

Results on burden of TRCs is one of important evidences to guide policy makers developing strategies on cancer prevention and control measures that are specific to consumption of tobacco smoking and use of smokeless tobacco, however it needs baseline scientific evidence in order to plan activities and evaluate the effectiveness of preventive protocols. The study aims to provide insightful baseline evidence on incidence of TRCs based on population-based cancer registries (PBCR) data and its distribution throughout diverse geographical locations of the country. The study will support national and provincial governments of Nepal to develop and modify tobacco-associated cancer prevention and control strategies for its effective and efficient implementation.

## Method and materials

Population-based cancer registry (PBCR) Nepal had collected data which was operated by Nepal Health Research Council (NHRC) since 1st January 2018 in different nine districts of the country including Kathmandu, Lalitpur, Bhaktapur from Bagmati province as representation of urban region, Siraha, Saptari, Dahnusha and Mohattari from Madesh province as representation of semi-urban and Rukum East from Lumbini province and Rukum West from Karnali province as representation of Rural PBCR. Urban, semi-urban and rural regions were categorized on the basis of availability of facilities like medical, educational and transportation etc. in the regions. Details about rural and urban PBCR of Nepal can be found in previous publication [14].

More than six million (6,249,759) population which (21% of total estimated population of the country) from urban, semi-urban and rural regions were covered by PBCR. Relevant data on cancer incidence were collected through health-facility based and community-based approaches using standard registry perfoma which includes the variables as recommended by IARC. Data from health facilities were collected by reviewing patient files, histopathology reports, case-summaries, reports of radio-diagnostics investigations (MRI, CT Scan etc.) and death certificates. On the other hand, trained data enumerators were mobilized in the communities to collect the data via face-face interview especially in semi-urban and rural regions as well as some rural parts of urban region in order to capture all incidence cases of defined geographical regions. Each patient or member of patient's family was briefed registration process and confidentiality while collecting data through face-to-face mode, and a separate consent form was signed for each primary data collection. Each collected case was reviewed for correctness and duplication, and verified by trained registry personnel before entry in CanReg5 software. The methodology of PBCR can also be found in report and articles published previously [12–14]. The Ethical Review Board (ERB) of NHRC has granted approval for this study.

For the purpose of this study, all cancer cases which are recognized to be tobacco related such as Cancer of lip (C00), tongue (C01–C02), mouth (C03–C06), oropharynx (C10), hypopharynx (C12–C13), pharynx (C14), esophagus (C15), larynx (C32), lung (C33–C34), and urinary bladder (C67) were used. We used PBCR data of mentioned topographies diagnosed from 1st January 2019 to 31st December 2019. Data on numbers, percent, age adjusted rate (AAR), age specific crude rate for each site, gender and geographical regions were computed using estimated population of the specified regions for year 2019 as well as world standard population to standardized the rate. Standard formula as defined by Day (1987) "{5*(summation of age specific rate of 0–74 years)}/100,000" for cumulative rate and "1-exp (-cumulative rate)" for cumulative risk respectively were used, and calculated using excel software [15]. The data related to cumulative risk of having tobacco-related cancer are presented in the clear and coherent way with the statement like one in number of individuals using "100/cumulative risk (%)" method which is one of the simplest epidemiological indicators to express the burden of disease.

## Findings

### Tobacco related cancer incidence

The study found that out of 3295 new cases 851 were tobacco related cancers. The proportion of TRCs among men and women was 35.3% and 17.3% respectively. The age adjusted rate of TRCs for men and women was 23.2 and 12.7 (per 100000 people) respectively. The overall cumulative lifetime risk (0–74 years) of tobacco-related cancers in Nepal revealed that 1 in 36 for men and 1 in 65 for women. The highest incidence of TRC for men (AAR 33.6 per 100,000) and women (AAR 17.0 per 100,000) was found in urban region as compared to semi-urban and rural regions which is shown in **Table 1**.

The highest incidence among men was observed for the lung cancer with AAR 9.8 per 100,000 population followed by mouth (AAR 4.1), bladder (AAR 2.6), larynx (AAR 2.0) and esophagus (AAR, 1.7) respectively. Similarly, in women, the leading TRCs were found the lung cancer with AAR 7.7 followed by tongue (AAR 1.2), esophagus (AAR 1.0), mouth (AAR 0.9), and larynx (AAR 0.7) respectively.

### Tobacco-related cancer deaths

Out of 1427 registered deaths cases 32.3% (men 41% and women 22.5%) were TRCs. Tobacco associated age adjusted cancer (AAR) mortality per 100,000 population was found 13.1 for men

**Table 1. Incidence of tobacco-related cancer versus all cancer in different regions of Nepal.**

| Region of Registry | Men | | | Women | | | Number of person likely to develop cancer in lifetime (0–74 years) | | | |
|---|---|---|---|---|---|---|---|---|---|---|
| | Number TRCs cases and percent (%) | AAR | | Number of TRCs cases and percent (%) | AAR | | Men | | Women | |
| | | TRC | All cancer | | TRC | All cancer | TRC | All cancer | TRC | All cancer |
| Urban | 339 (36.1%) | 33.6 | 86.7 | 187 (17.3%) | 17 | 90.8 | 1 in 24 | 1 in 10 | 1 in 47 | 1 in 10 |
| Semi-Urban | 199 (34.4%) | 16.5 | 47.8 | 99 (16.2%) | 8.7 | 49 | 1 in 50 | 1 in 21 | 1 in 95 | 1 in 18 |
| Rural | 13 (30.3%) | 16.6 | 52.5 | 14 (31.1%) | 15.3 | 47.3 | 1 in 50 | 1 in 17 | 1 in 67 | 1 in 21 |
| All regions | 551 (35.3%) | 23.2 | 63.5 | 300 (17.3%) | 12.7 | 68.1 | 1 in 36 | 1 in 14 | 1 in 65 | 1 in 13 |

**Note:** AAR denotes Age Adjusted Rate (world) per 100,000 population and % is relative proportion of all cancer cases registered in 2019

and 6.3 for women. Among TRCs mortality. lung cancer was the major cause of death which comprised 24.1% (AAR 7.9 per 100,000) of total death among men followed by mouth 4.7% (AAR 1.5), esophagus 3.2% (AAR 1.0), larynx 2.4% (AAR 0.8) and bladder 1.8% (AAR 0.6).

Similarly, the lung cancer contributed the highest proportion of death in women as well with 17.1% (AAR 4.9) followed by esophagus 1.3% (AAR 0.3), mouth and tongue cancer each comprised 1% (AAR 0.3)., Lung cancer was the commonest cause of death identified in all regions with higher proportion in rural (men 34.8% and women 17%) and urban regions (men 33.1% women 22.6%) compared to semi-urban region (men 10.4% and women 8.3%). Death from mouth cancer was found high in semi-urban regions (men 8.7% and women 2.9%) of the country.

## Cumulative risk (lifetime) of having cancer (0–74 years)

The risk of having lung cancer was the highest in both sex with 1 in 79 men and 1 in 106 women followed by mouth cancer (1 in 216) in men and tongue cancer in women with the risk of 1 in 613 women. The development of cancer of oropharynx and pharynx were identified as the lowest risk in Nepal with 1 in 3,425 men and 1 in 17,857 women respectively.

The risk for developing lung cancer was highest in urban region (1 in 45 men and 1 in70 women) followed by rural region (1 in 61 men and 1 in 68 women). Similarly, men in semi-urban region have greatest risk for developing mouth cancer (1 in 141 men) followed by lung (1 in 174), and in women the risk was vice-versa with 1 in 199 for lung and 1 in 612 for mouth cancer. The number of TRC cases, proportion of TRC with all cancer, age adjusted rate (AAR), and cumulative risk percentage for each TRC sites for men and women of each PBCR locations is presented in **Tables 2 and 3** respectively (Tables 2 and 3).

## Discussion

The study finding showed that more than one third (35.3%) among men and less than one fifth (17.3%) of all cancer among women were tobacco-related. There was slight reduction in the overall proportion of TRCs in 2019 compared to 2018 (men 36.3% and women 14.7%) in Nepal this may be due to under-reporting of cancer cases; however, ongoing surveillance will provide further stability on cancer burden. The scientific community still lacks the supporting evidence on tobacco-related cancers in Nepal except one hospital-based study which was conducted within a health facility in western Nepal that showed higher proportion (men 48% and women 28%) of TRCs in both genders [16]. The proportion of TRCs in different regions of India range from 11–25% for men and 3–18% for women, and the overall share of TRCs in India was 45% and 20% for men and women respectively which was higher compared to Nepal [11]. The PBCR of Varanasi in India reported the highest burden of TRCs in India. The TRCs burden in Nepal for men is low as compared to Varanasi and other Indian registries such as Sangrur, Chandigarh, Delhi, Barshi rural, Mumbai and Chennai as well as other registries in United Kingdom, United States and Curitiba, Brazil. However, the burden of TRCs for women in Nepal is higher as compared to Varanasi and above-mentioned registries [17]. Tobacco-related cancers in Muzaffarpur district (a district of Bihar province, India which share border with semi-urban PBCR region of Nepal) among men was 36.5% which was almost similar with the TRCs among men in semi-urban region (SSDM registry) of Nepal. However, the proportion of TRCs among women (8.8%) in Muzaffarpur was almost 50% lower compared to women in Nepal [18].

Approximately one third (32.3%) of all cancer deaths (men 41% and women 22.5%) was identified to be associated with tobacco in Nepal. The share of death associated to be TRCs among men in Nepal was almost comparable with the results in India where 42.0% men and

**Table 2. Incidence of tobacco-related cancers in different geographical regions of Nepal, Men (N = 1559).**

| PBCR region | | Lip (C00) | Tongue (C01-02) | Mouth (C03-06) | Oropharynx (C10) | Hypopharynx (C12-13) | Pharynx (C14) | Esophagus (C16) | Larynx (C32) | Lungs (C33-34) | Bladder (C67) |
|---|---|---|---|---|---|---|---|---|---|---|---|
| Urban | N | 3 | 20 | 26 | 7 | 12 | 2 | 27 | 34 | 165 | 43 |
| | % | 0.3 | 2.1 | 2.8 | 0.7 | 1.3 | 0.2 | 2.9 | 3.6 | 17.6 | 4.6 |
| | AAR | 0.3 | 1.9 | 2.4 | 0.6 | 1.2 | 0.2 | 2.6 | 3.5 | 16.7 | 4.2 |
| | Cum Risk % (*) | 0.0277 (3610) | 0.2164 (462) | 0.2454 (407) | 0.0694 (1441) | 0.1633 (612) | 0.0575 (1739) | 0.3197 (313) | 0.4698 (213) | 2.2402 (45) | 0.4875 (205) |
| Semi-urban | N | 7 | 15 | 75 | - | 5 | 4 | 13 | 11 | 54 | 15 |
| | % | 1.2 | 2.6 | 13 | - | 0.9 | 0.7 | 2.2 | 1.9 | 9.3 | 2.6 |
| | AAR | 0.5 | 1.2 | 6.2 | - | 0.4 | 0.3 | 1.2 | 1.0 | 4.4 | 1.3 |
| | Cum Risk % (*) | 0.0879 (1138) | 0.1321 (757) | 0.7074 (141) | - | 0.0607 (1647) | 0.036 (2778) | 0.1092 (916) | 0.114 (877) | 0.5746 (174) | 0.1314 (761) |
| Rural | N | - | - | - | - | - | - | - | 2 | 8 | 3 |
| | % | - | - | - | - | - | - | - | 4.7 | 18.6 | 7 |
| | AAR | - | - | - | - | - | - | - | 2.4 | 9.7 | 4.4 |
| | Cum Risk % (*) | - | - | - | - | - | - | - | 0.1308 (765) | 1.6475 (61) | 0.254 (394) |
| All regions | N | 10 | 35 | 101 | 7 | 17 | 6 | 40 | 47 | 227 | 61 |
| | % | 0.6 | 2.2 | 6.5 | 0.4 | 1.1 | 0.4 | 2.6 | 3.0 | 14.6 | 3.9 |
| | AAR | 0.4 | 1.4 | 4.1 | 0.3 | 0.7 | 0.2 | 1.7 | 2.0 | 9.8 | 2.6 |
| | Cum Risk % (*) | 0.0614 (1629) | 0.15 36 (651) | 0.4639 (216) | 0.0292 (3425) | 0.1021 (979) | 0.0400 (2500) | 0.1843 (543) | 0.2523 (396) | 1.2647 (79) | 0.2719 (368) |

Note: N-Number of incidence cases, %- relative proportion of TRCs among all cases, AAR- age adjusted rate per 100,000 population and (*) denotes one in number of person (0–74 Years) likely to develop cancer in lifetime.

18.3% women death was due to tobacco use [19], though, the proportion of smoking related death in Chinese people (28.87% among men and 13.37% among women) was considerably lower compared to Nepal [20].

Every one in 80 men and 107 women are at risk of getting lung cancer before age of 75 in Nepal, however, the risk for lung cancer in India (1 in 67 person) was less compared to Nepal [11]. Lung cancer is the commonest cause of tobacco associated cancer in most developing countries including China, and also in developed country like the United States [20–22]. We found lung cancer the common as well as the most fetal cancer in Nepal, which has been remained same since long [23–25]. The proportion of lung cancer among women of rural region was increased significantly from 9.3% in 2018 to 22.2% in 2019 [12, 26] this may be due to improvement in death registration. The most important known cause of lung cancer is tobacco smoking which contain majority of human carcinogens [6] and long term exposure to pollution [27]. Overall 17.1% people smoke tobacco in Nepal [10]. Increase in incidence of lung cancer may be associated with the long term daily exposure to smoke from unprocessed biomass fuels produced during burning of wood, dung, and crop residues. Scholar reported that people in developing countries mostly in rural household especially women are commonly exposed to high levels of indoor pollution for 3–7 hour everyday over many years for their daily work (i.e. Cooking) [28].

Smokeless tobacco product such as oral snuff, chewing tobacco (Gutkha, Khaini, Surti, Supari/Areca nuts and Pan/Betel leaf etc.) contain relatively low level of carcinogens compared

**Table 3. Incidence of tobacco-related cancers in different geographical regions of Nepal, Women (N = 1736).**

| PBCR regions | | Lip (C00) | Tongue (C01-02) | Mouth (C03-06) | Oropharynx (C10) | Hypopharynx (C12-13) | Pharynx (C14) | Esophagus (C16) | Larynx (C32) | Lungs (C33-34) | Bladder (C67) |
|---|---|---|---|---|---|---|---|---|---|---|---|
| Urban | N | 3 | 17 | 7 | 1 | 3 | - | 19 | 7 | 122 | 8 |
| | % | 0.3 | 1.6 | 0.6 | 0.1 | 0.3 | - | 1.8 | 0.6 | 11.3 | 0.7 |
| | AAR | 0.2 | 1.7 | 0.6 | 0.1 | 0.2 | - | 1.6 | 0.8 | 11 | 0.7 |
| | Cum Risk % (*) | 0.0313 (3195) | 0.2387 (419) | 0.497 (2012) | 0.0051 (19608) | 0.0224 (4464) | - | 0.1978 (506) | 0.1113 (898) | 1.4199 (70) | 0.0820 (1220) |
| Semi-urban | N | - | 11 | 14 | 3 | 1 | 1 | 5 | 9 | 49 | 6 |
| | % | - | 1.8 | 2.3 | 0.5 | 0.2 | 0.2 | 0.8 | 1.5 | 8.0 | 1.0 |
| | AAR | - | 1.0 | 1.2 | 0.3 | 0.1 | 0.1 | 0.4 | 0.8 | 4.4 | 0.5 |
| | Cum Risk % (*) | - | 0.1185 (844) | 0.1633 (612) | 0.0435 (2299) | 0.0151 (6623) | 0.0100 (10000) | 0.0681 (1468) | 0.0723 (1383) | 0.5037 (199) | 0.0623 (1605) |
| Rural | N | - | 1 | - | - | - | 1 | - | 1 | 10 | 1 |
| | % | - | 2.22 | - | - | - | 2.22 | - | 2.22 | 22.2 | 2.22 |
| | AAR | - | 0.6 | - | - | - | 1.3 | - | 1.3 | 10.9 | 1.3 |
| | Cum Risk % (*) | - | 0.0316 (1365) | - | - | - | - | - | - | 1.475 (68) | - |
| All regions | N | 3 | 29 | 21 | 4 | 4 | 2 | 24 | 17 | 181 | 15 |
| | % | 0.2 | 1.7 | 1.2 | 0.2 | 0.2 | 0.1 | 1.4 | 1.0 | 10.4 | 0.9 |
| | AAR | 0.1 | 1.2 | 0.9 | 0.2 | 0.2 | 0.1 | 1.0 | 0.7 | 7.7 | 0.6 |
| | Cum Risk % (*) | 0.0150 (6667) | 0.1631 (613) | 0.1024 (977) | 0.0250 (4000) | 0.0180 (5556) | 0.0056 (17857) | 0.1220 (820) | 0.0854 (1171) | 0.9396 (106) | 0.0684 (1462) |

Note: N-Number of incidence cases, %- relative proportion of TRCs among all cases, AAR- age adjusted rate per 100,000 population and (*) denotes one in number of person (0–74 Years) likely to develop cancer in lifetime.

to smoked tobacco, however, the Nitrosamines, the most prevalent strong carcinogen in unburned tobacco are known to be responsible for causing cancer of oral cavity, pancreas, esophagus etc [6, 29, 30]. Overall, 18.3% of current tobacco user in Nepal use smokeless tobacco product, and two third (71.4%) of them use surti (dried tobacco leaves) or khaini (lime missed tobacco). The men to women ratio of smokeless tobacco user in Nepal was 6.7 [10]. Consequently, cancer of oral cavity (including lip, tongue, mouth and pharynx) in Nepal was found significantly higher in men (11.2%) compared to women (3.6%) which was comparable with the results found in Sri Lanka [31] as well as Europe [32]. Every one in 216 men and 1 in 977 women is at risk for mouth cancer in Nepal which was likely to be similar with the finding in India that shows the overall risk of mouth cancer was 1 in 250 in the southern and central regions of the country [11].

Every 1 in 553 men and 1 in 820 women were at risk for esophageal cancer in Nepal, it was significantly lower compared to 1 in 27 people in India especially in Northeastern regions [11], although, scholars reported high prevalence of chewing Areca nuts (plain Supari) in Nepal which was found associated with oral and esophageal cancer [33, 34].

Nepal has been implementing operational policies and action plan since more than a decade through Tobacco Product Control Act 2011, WHO FCTC and MPOWER measures which includes making all public places smoke-free; providing large, legible warning labels; increasing tax on tobacco products as well as banning advertisement on national television, radio, and print media etc. Despite of the operation of such tobacco control programs,

tobacco-related cancer is high in Nepal, and the proportion of current smokers who tried to quit tobacco use was significantly decreased from 26.1% in 2013 to 19.4% in 2019. In addition, mean age of initiating tobacco smoking in Nepal remains same since 2013 [10]. Effective implementation of behavioral counselling as well as quit tobacco program, more than half of current tobacco smokers (55.4%) and almost half of smokeless tobacco user (49.6%) in India were thinking to quit tobacco which was almost two fold higher in compared to Nepal [35].

The result suggests to focus on monitoring of tobacco control strategies and its effectiveness along with the introduction of new effective educational and awareness modalities such as behavioral counseling and active follow up which can help the quitters to quit tobacco successfully. Group or individual psychological support through behavioral counseling using the steps 5 A'S (Ask about tobacco use, Assess to quit, Assess commitment and barrier to change, Assist user committed to change and Arrange follow-up to monitor progress) to those who were willing to quit is cost effective approach to reduce habit-associated ill health. Furthermore, educational advocacy and behavioral counseling using 5 R's model (Relevance, Risk, Rewards, Roadblocks and Repetition) can help tobacco user to quit tobacco successfully although they were not willing to quit tobacco [36]. Cancer awareness through school education with structured session planning including early signs, symptoms, risks factors, preventive measures (such as stopping smoking and tobacco use), and treatment modalities helps to promote disease awareness and support early detection [37]. It is recommended that Tobacco Quitline Services for tobacco control should be started in Nepal as these services are beneficial in India [38, 39].

Last but not the least, authors want to explore some limitation of the present study that the study provides tobacco-related cancer statistics in all geographical regions of the country; however, it does not cover entire population of the regions because of limited population coverage by PBCR. Moreover, PBCR Nepal does not registered information on tobacco use and other risk factors. Thus, the findings was analyzed based on the site of cancer that are known to be associated with tobacco use, however, it does not estimate the proportion of cancer directly caused by tobacco use. Beside that, correct and consistent data collection as well as use of principles and guidelines of International Association of Cancer Registries (IACR) by PBCR Nepal will provide robust baseline evidence on TRCs burden of the country.

## Conclusion

More than one third of all cancer in all geographical regions were tobacco associated in Nepal., Government of Nepal should give priority to strong implementation of tobacco control strategies and perform periodic monitoring of its execution. There should be a tobacco quit helpline that provide behavioral counseling to individuals who try to quit. Population-based cancer registries should be on government priority and expanded to all provinces to monitor the burden of cancer in the country. Finally, we recommend further studies on TRCs in relation to tobacco use and its association with age and other socio-demographic factors to provide more consistent data in this regard.

## Supporting information

**S1 Checklist. Human participants research checklist.**
(DOCX)

## Acknowledgments

We would especially want to thank the Secretary of the Ministry of Health and Population (MoHP), Government of Nepal, for their invaluable support and encouragement in helping

Nepal to initiate the Population-Based Cancer Registry (PBCR). We appreciate the continuous technical assistance for data quality monitoring and capacity building provided by the International Agency for Research on Cancer (IARC) Regional Hub at Tata Memorial Center, Mumbai, India and the WHO Country Office Nepal. Our deepest appreciation goes to the Health Tax Fund under ministry of health and population for their continuous support to sustain the registry in Nepal. Thank you to everyone who helped establish the PBCR in the urban, semi-urban and rural setting of the country including the local government, cancer treatment and diagnostic facilities, hospices social security and nursing division under the department of health services. Finally, we would like to thank all registry staffs for their dedication and effort to run PBCR in Nepal.

## Author Contributions

**Conceptualization:** Uma Kafle Dahal, Meghnath Dhimal, Gehanath Baral, Anjani Kumar Jha, Sandhya Chapagain.

**Data curation:** Uma Kafle Dahal, Meghnath Dhimal, Kopila Khadka, Sudha Poudel.

**Formal analysis:** Uma Kafle Dahal, Meghnath Dhimal, Atul Budukh.

**Methodology:** Uma Kafle Dahal.

**Project administration:** Uma Kafle Dahal, Meghnath Dhimal, Kopila Khadka, Pradip Gyanwali, Anjani Kumar Jha.

**Supervision:** Anjani Kumar Jha, Sandhya Chapagain.

**Validation:** Meghnath Dhimal, Atul Budukh, Kopila Khadka, Gehanath Baral.

**Writing – original draft:** Uma Kafle Dahal.

**Writing – review & editing:** Meghnath Dhimal, Atul Budukh, Sudha Poudel, Gehanath Baral, Pradip Gyanwali, Anjani Kumar Jha, Sandhya Chapagain.

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
