## [Decision Letter · Decision Letter 0]

13 Dec 2023

PONE-D-23-32604Burden of tobacco-related cancers in urban, semi-urban and rural setting of Nepal: results from population-based cancer registries 2019PLOS ONE

Dear Dr. KAFLE DAHAL,

Thank you for submitting your manuscript to PLOS ONE. After careful consideration, we feel that it has merit but does not fully meet PLOS ONE’s publication criteria as it currently stands. Therefore, we invite you to submit a revised version of the manuscript that addresses the points raised during the review process.

**ACADEMIC EDITOR: Please insert comments here and delete this placeholder text when finished.** Be sure to:

1. Though cigarette consumption is decreasing in Nepal, smokeless tobacco users are increasing in Nepal. How will your paper contribute to reducing such a burden? Explain introduction. 

2. The method needs to be extended. It means providing in detail. The time period of Data collection is not clear. 

3. There is no consistency in writing. For Example, In the text, it is written male and female while in the table it is men and women.

4. Please provide the number of male and female participants in the table so the reader can understand the % in the table. Also, provide footnotes for explaining %. 

5. Authors have mentioned limitations but not strengths of the   

   study. Please mention it. 

6. The definition of Urban-rural and semi-urban areas is not clear. 

7. The calculation of cumulative risk is not clear. Please provide Step-by-step methods.

8. 0-79 years. Can you clarify on this age group? How 0 years is related to TRC?

9. Why authors focused on place of residence but not age and other demographic factors? ( See tables)

10. It is not clear how incidence rate were computed in this study?

Please submit your revised manuscript by Jan 27 2024 11:59PM. If you will need more time than this to complete your revisions, please reply to this message or contact the journal office at plosone@plos.org. Please include the following items when submitting your revised manuscript:A rebuttal letter that responds to each point raised by the academic editor and reviewer(s). You should upload this letter as a separate file labeled 'Response to Reviewers'.A marked-up copy of your manuscript that highlights changes made to the original version. You should upload this as a separate file labeled 'Revised Manuscript with Track Changes'.An unmarked version of your revised paper without tracked changes. You should upload this as a separate file labeled 'Manuscript'.

We look forward to receiving your revised manuscript.

Kind regards,

Umesh Raj Aryal, PhD

Academic Editor

PLOS ONE

Journal Requirements:

Additional Editor Comments :

1. Though cigarette consumption is decreasing in Nepal, smokeless tobacco users are increasing in Nepal. How will your paper contribute to reducing such a burden? Explain introduction.

2. The method needs to be extended. It means providing in detail. The time period of Data collection is not clear.

3. There is no consistency in writing. For Example, In the text, it is written male and female while in the table it is men and women.

4. Please provide the number of male and female participants in the table so the reader can understand the % in the table. Also, provide footnotes for explaining %.

5. Authors have mentioned limitations but not strengths of the

study. Please mention it.

6. The definition of Urban-rural and semi-urban areas is not clear.

7. The calculation of cumulative risk is not clear. Please provide Step-by-step methods.

8. 0-79 years. Can you clarify on this age group? How 0 years is related to TRC?

9. Why authors focused on place of residence but not age and other demographic factors? ( See tables)

10. It is not clear how incidence rate were computed in this study?

Reviewers' comments:

Reviewer's Responses to Questions

**Comments to the Author**

1. Is the manuscript technically sound, and do the data support the conclusions?

Reviewer #1: Yes

Reviewer #2: Yes

Reviewer #3: Yes

2. Has the statistical analysis been performed appropriately and rigorously? 

Reviewer #1: Yes

Reviewer #2: I Don't Know

Reviewer #3: I Don't Know

3. Have the authors made all data underlying the findings in their manuscript fully available?

Reviewer #1: Yes

Reviewer #2: Yes

Reviewer #3: Yes

4. Is the manuscript presented in an intelligible fashion and written in standard English?

Reviewer #1: Yes

Reviewer #2: No

Reviewer #3: Yes

5. Review Comments to the Author

Reviewer #1: The submitted paper has important information in the policy making level. The author need to address the following points:

- Title: Instead of writing result from the population based..... it would be good to write findings.

- Introduction: Delete space between reference and full stop at Line 30. In the title you have mentioned burden of tobacco however there is no any literature supporting point in introduction and rational

- Methodology: Recruitment criteria are not clearly defined, what was the sampling technique? It would be good to write the operational definition of Urban and semi-urban.

Results: Data with demographic details and the prevalence with age, region, factors would be very helpful for the readers.

Discussion: Two sentence from174 to 176 are not clear. Lung cancer in Nepal.....

Reviewer #2: Queries

- Can you mention the sample age group in the methodology section?

- What are the exclusion criteria in the study?

- Can you mention the study site and study time? How many hospitals were involved?

- Were there any biases in the secondary data?

- How did you differentiate the data for rural and urban regions?

- How can you be sure to generalize the result?

-

Comments

- Citations are missing in some parts (for example, in the discussion)

- There are lots of grammatical, typographical, and syntax errors.

- Uppercase and lowercase need to be corrected including punctuation.

- Lots of abbreviations and full forms need to be addressed. (for example, full form for NCDs, SSDM, GLOBOCON, etc)

- Sentences are not clear and need to be restructured. (for example, line 129, etc)

Reviewer #3: Line number 19 in abstract section, lung, mouth, esophageal and larynx cancer are the top 4 cancer, NOT TOP 5 TRC in Nepal.

Please mention what mouth includes in your paper(lips,tongue,buccal cavity), this will clarify the readers. Does mouth cancer also include tongue cancer in a whole?

Line 29, it would be nice to add predictions for upcoming years rather than past years.

line 99, Please arrange the sentence and make it more meaningful to the readers.

Please clarify what tobacco related cancer means in introduction? What form of tobacco use does TRC include? If it includes smoking and oral forms both, then do mention it.

6. PLOS authors have the option to publish the peer review history of their article (what does this mean?). If published, this will include your full peer review and any attached files.

Reviewer #1: **Yes: **Bishnu Dutta Acharya

Reviewer #2: No

Reviewer #3: No

---

## [Author Response · Author response to Decision Letter 0]

4 Jan 2024

We have addressed all the concern and queries, and the manuscript has been revised and uploaded as requested. A separate rebuttal letter also uploaded in the system. 

Thank you

Uma Kafle Dahal

---

## [Decision Letter · Decision Letter 1]

14 Feb 2024

PONE-D-23-32604R1Burden of tobacco-related cancers in urban, semi-urban and rural setting of Nepal: findings from population-based cancer registries 2019PLOS ONE

Dear Dr. KAFLE DAHAL,

Thank you for submitting your manuscript to PLOS ONE. After careful consideration, we feel that it has merit but does not fully meet PLOS ONE’s publication criteria as it currently stands. Therefore, we invite you to submit a revised version of the manuscript that addresses the points raised during the review process.

We look forward to receiving your revised manuscript.

Kind regards,

Umesh Raj Aryal, PhD

Academic Editor

PLOS ONE

Journal Requirements:

Additional Editor Comments:

needs to address further comments by reviewers

Reviewers' comments:

Reviewer's Responses to Questions

**Comments to the Author**

1. If the authors have adequately addressed your comments raised in a previous round of review and you feel that this manuscript is now acceptable for publication, you may indicate that here to bypass the “Comments to the Author” section, enter your conflict of interest statement in the “Confidential to Editor” section, and submit your "Accept" recommendation.

Reviewer #4: (No Response)

Reviewer #5: All comments have been addressed

2. Is the manuscript technically sound, and do the data support the conclusions?

Reviewer #4: Yes

Reviewer #5: Yes

3. Has the statistical analysis been performed appropriately and rigorously? 

Reviewer #4: Yes

Reviewer #5: Yes

4. Have the authors made all data underlying the findings in their manuscript fully available?

Reviewer #4: No

Reviewer #5: Yes

5. Is the manuscript presented in an intelligible fashion and written in standard English?

Reviewer #4: No

Reviewer #5: Yes

6. Review Comments to the Author

Reviewer #4: 1. Please add research gaps at the end of the introduction.

2. Methods need to be elaborated:

a. Sampling (size and Techniques)

b. Data collection instruments

c. Maintenance of Ethical Issues on aspect of the respondents.

d. Add operational definitions of urban, semi-urban and rural.

e. Add dependent and independent variables.

f. Add patient identification time from the registry.

3. Results:

a. Add table or figure number after the paragraph of result section.

b. Make footnote regarding the short text and analysis technique mentioned ion the tables.

4. Add strengths of the study.

Reviewer #5: The study is completely secondary and different parts from study has been published earlier. Authors has well presented the findings. Manuscript can be published based on publication criteria of Plos One.

7. PLOS authors have the option to publish the peer review history of their article (what does this mean?). If published, this will include your full peer review and any attached files.

Reviewer #4: **Yes: **Dr. Bilkis Banu

Reviewer #5: No

---

## [Author Response · Author response to Decision Letter 1]

21 Feb 2024

Dear Editors and Reviewers

Warm greetings!

I am grateful for your time to review the manuscript and insightful comments which support us to make manuscript more meaningful. In response to your email requiring revision, I went through the manuscript as well as the list of reference. I have checked it again for its completeness and correctness to each article cited in our manuscript using Pub med and google scholars database as well as on the latest version of ZOTERO as well. I found none of the articles cited in our manuscript has been retracted. I would be grateful if you support us to identify any articles cited in our manuscript deemed to be confused. 

I have changed source of information for line 33 (Reference no.2), and did some revision in the manuscript as suggested by reviewer which has been highlighted in tract change mode.

In response to the concern raised by Editor #4

First of all, I would like to thank you for your insightful revision on the manuscript. As the data collected by population-based cancer registry. It is always a census based where each and every cancer cases diagnosed in the specified period will be registered. Data was collected using structured questionnaires which includes variables recommended by International Association of cancer registries (IACR). 

Concern about ethical consideration, secondary data was collected from health facilities taking prior permission from concerned authorities, and all the ethical consideration including individual consent taking was applied while collecting primary data through face-to-face method from patient or his/her family members. I have mentioned about the ethical consideration practices we followed during data collection. 

Urban, semi-urban and rural regions was categories on the basis of geographical locations, and availability of facilities in the regions which was further clarified in our previous publication (reference no.14). I have done revision addressing the suggestion received from reviewer.

Thank you

Uma Kafle Dahal

(on behalf of all authors)

---

## [Editor Report · Decision Letter 2]

26 Feb 2024

Burden of tobacco-related cancers in urban, semi-urban and rural setting of Nepal: findings from population-based cancer registries 2019

PONE-D-23-32604R2

Dear Kafle,

We’re pleased to inform you that your manuscript has been judged scientifically suitable for publication and will be formally accepted for publication once it meets all outstanding technical requirements.

Kind regards,

Umesh Raj Aryal, PhD

Academic Editor

PLOS ONE

Additional Editor Comments (optional):

None

Reviewers' comments:

None

---

## [Editor Report · Acceptance letter]

15 May 2024

PONE-D-23-32604R2 

PLOS ONE

Dear Dr. KAFLE DAHAL, 

I'm pleased to inform you that your manuscript has been deemed suitable for publication in PLOS ONE. Congratulations! Your manuscript is now being handed over to our production team.

Kind regards, 

on behalf of

Dr. Umesh Raj Aryal 

Academic Editor

PLOS ONE